# DualEdit: Dual Editing for Knowledge Updating in Vision-Language Models

**Zhiyi Shi**[1]*, **Binjie Wang**[1,4]*, **Chongjie Si**[5], **Yichen Wu**[1,2]†‡, **Junsik Kim**[1,3]†, **Hanspeter Pfister**[1]
*Equal contribution, ‡Project leader, †Corresponding author
[1]Harvard University, [2]City University of Hong Kong, [3]Amazon, [4]Fudan University
[5]MoE Key Lab of Artificial Intelligence, AI Institute, Shanghai Jiao Tong University
{gabshi59, bjwang02.hz, wuyichen.am97, mibastro}@gmail.com

## Abstract

Model editing aims to efficiently update a pre-trained model's knowledge without the need for time-consuming full retraining. While existing pioneering editing methods achieve promising results, they primarily focus on editing single-modal language models (LLMs). However, for vision language models (VLMs), which involve multiple modalities, the role and impact of each modality on editing performance remain largely unexplored. To address this gap, we explore the impact of textual and visual modalities on model editing and find that: (1) textual and visual representations reach peak sensitivity at different layers, reflecting their varying importance and (2) editing both modalities can efficiently update knowledge, but this comes at the cost of compromising the model's original capabilities. Based on our findings, we propose DualEdit, an editor that modifies both textual and visual modalities at their respective key layers. Additionally, we introduce a gating module within the more sensitive textual modality, allowing DualEdit to efficiently update new knowledge while preserving the model's original information. We evaluate DualEdit across multiple VLM backbones and benchmark datasets, demonstrating its superiority over state-of-the-art VLM editing baselines as well as adapted LLM editing methods on different evaluation metrics. Codes are available at https://github.com/zhiyiscs/DualEdit.

## 1 Introduction

Large Language Models (LLMs) and Vision Language Models (VLMs) have achieved remarkable success across a wide range of natural language processing tasks (Dong et al., 2024; Sheng et al., 2024; Zhu et al., 2023b; Yin et al., 2024), demonstrating strong capabilities in reasoning, knowledge retrieval, and text generation. Despite these advancements, the knowledge encapsulated within LLMs and VLMs usually becomes static after training, which makes it difficult for them to update and correct errors, incorporate new knowledge, or refine specific behaviors in real-world applications (Liang et al., 2024) . To efficiently alleviate this problem, model editing has emerged as a promising solution (Wang et al., 2023; Hartvigsen et al., 2023; Chen et al., 2024a; Jiang et al., 2024), allowing targeted modifications to a model's predictions while preserving overall performance and minimizing unintended changes to unrelated inputs.

Previous model editing methods have developed many efficient strategies to address this problem. For instance, methods like ROME (Meng et al., 2022a), MEND (Mitchell et al., 2021), and MEMIT (Meng et al., 2022b) achieve knowledge edits by applying offsets to specific model parameters, while memory-based methods like SERAC (Mitchell et al., 2022), GLAME (Zhang et al., 2024a), and WISE (Wang et al., 2024) leverage external memory for targeted edits. However, most current model editing methods are designed for single-modal models and cannot easily adapt to the increasing significance of multi-modal models (Liang et al., 2024), such as VLMs. As pointed out by Cheng et al. (2023), who constructed a

multimodal editing benchmark, traditional single-modal editing methods perform poorly in multimodal scenarios. This challenge arises because errors in multimodal models frequently stem from the complex interactions between different modalities, such as the intertwined influence of both visual and textual modalities in VLMs (Liang et al., 2024) . Therefore, to effectively address the complexities of multi-modal model editing, it is crucial to first *fully explore the specific role of each modality and their key layers*.

In spite of the increasing prominence of VLMs, existing research on editing these multimodal models is still limited (Cheng et al., 2023). Focusing on editing in VLMs, VisEdit (Chen et al., 2024b) is the first work to identify key layers in the visual modality and edit them to update the model's knowledge. However, while achieving good performance, they concentrate solely on the visual modality and completely neglect the textual modality, which fails to fully recognize the roles of different modalities in VLMs.

To gain a deep understanding of the roles different modalities play in knowledge editing for VLMs and to offer valuable insights for designing multi-modal knowledge editing methods, we begin by conducting a thorough empirical analysis of each modality's contribution to the model's overall performance. Specifically, through carefully designed experiments, we explore and analyze from the following two perspectives:

- **Layer-wise and modality-wise importance.** In order to investigate the sensitivity of visual and textual modalities, we first analyze the attention scores of different modalities at various layers, which reflect the relative importance of each modality at each layer. The results reveal that, within the same layer, textual modalities receive higher attention scores than visual modalities. Combining this with experiments on the impact of perturbations to each modality at different layers on the final performance, we conclude that the importance of modalities differs both across layers and within the same layer. Thus, it is crucial to address each modality separately at every layer during knowledge editing.

- **Trade-Off between reliability and locality.** By directly editing visual and textual modalities, we find that they can easily adapt to injected modifications for updating new knowledge (i.e., improving the Rel. performance), but at the same time, it also faces a greater risk of disrupting existing knowledge (i.e., decreasing the Loc. metric), which aligns with Gekhman et al. (2024). Consequently, while considering the importance of different modalities, we must also be mindful of the trade-off between reliability and locality when editing specific modalities.

Based on these two findings, we introduce DualEdit, a novel editing approach for VLMs that takes into account the distinct effects of textual and visual modalities during editing. Unlike conventional editing methods that treat multimodal inputs uniformly, DualEdit performs modality-aware modifications by applying edits at different layers for visual and textual features. Specifically, we design a gating module that selectively decides whether to edit a model's response using a learnable adapter. By applying this mechanism separately to the textual and visual modalities, we enable modality-specific editing in VLMs while ensuring a well-balanced trade-off between reliability and locality. To sum up, our main contributions can be summarized as follows:

- We are the first to conduct comprehensive experiments that decouple the analysis of different modalities in VLM knowledge editing, examining their impact and identifying their relative and absolute importance both across layers and within individual layers.
- Based on our findings, we further propose DualEdit, a modality-aware editing approach that operates on some key layers, ensuring a well-balanced trade-off between reliability and locality by incorporating the designed gating module.
- We conduct comprehensive quantitative and ablation experiments across multiple VLM backbones and benchmark datasets, demonstrating the superiority of DualEdit over state-of-the-art VLM editing baselines as well as adapted LLM editing methods.

## 2   Analysis of Different Modalities in VLMs

Most existing model editing methods primarily focus on single-modality in LLMs. However, the lack of a comprehensive analysis across different modalities has limited the development

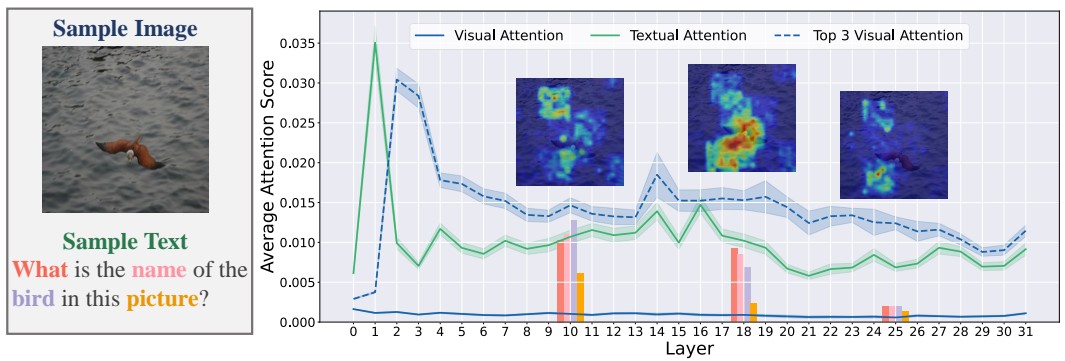

Figure 1: The average attention scores of textual representations and visual representations across different layers in LLaVA-V1.5 (Liu et al., 2023a). The dashed line illustrates the mean attention scores of the three visual representations receiving the highest attention values in each layer. The right figure also shows the attention scores of the sample at [10, 18, 25] layers.

of effective model editing techniques for VLMs. To better understand these impacts, we randomly selected 1000 samples and conducted exploratory experiments across various layers using the widely adopted LLaVA-V1.5 backbone. The results can be summarized in the following two key findings:

Finding-1: *Textual and Visual representations in VLMs demonstrate varying levels of importance within the same layer, and the significance of each modality changes across different layers.* As shown in Figure 1, the average attention scores of textual modalities are significantly higher than those of visual modalities, with only a few visual tokens (e.g., the top 3 visual tokens) exhibiting high scores, as indicated by the dashed line. Moreover, the attention scores of both modalities peak at different shallow layers. These observations both highlight the distinct treatment of textual and visual modalities across the layers of VLMs, underscoring the necessity of handling different modalities separately.

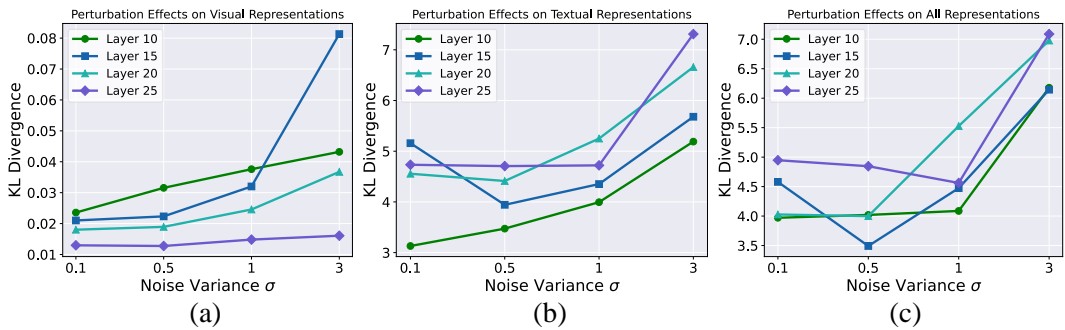

Figure 2: KL divergence between original and perturbed output logits when adding gaussian noise to (a) visual, (b) textual, and (c) all representations at different layers with different noise variance $\sigma$.

On the other hand, to explore the absolute importance of each modality at different layers of VLMs, we introduce varying levels of Gaussian noise into specific layers of a given modality and analyze the resulting changes in the output, as depicted in Figure 2 (a)(b). It can be observed that perturbations at different layers affect the textual and visual modalities differently, indicating that the importance of each layer varies across modalities. In addition to applying perturbations separately, we also present the results of perturbing both modalities simultaneously, as shown in Figure 2 (c). This reveals a distinct pattern compared to previous cases, suggesting that the influence of both modalities on the output is not simply additive. This further underscores the need to carefully consider the interactions between modalities when designing editing methods for VLMs. In summary, by combining the

results of Figure 1 and Figure 2, we can summarize Finding 1, where the important layers of both modalities lie in different layers.

**Finding-2:** *In VLMs, editing textual and visual modalities can improve the performance of edited samples, but it significantly impacts the performance on original samples (i.e., locality performance).*

As illustrated in Table 1, we perform editing on different textual layers (i.e., T-Layer) and visual layers (i.e., V-Layer). The results show that, compared to the visual modality, the textual modality adapts more easily to edited samples, achieving higher Rel. performance. However, modifying either the textual or visual modality at different layers negatively impacts the model's original performance, leading to lower Loc. performance. This underscores the importance of designing effective editing strategies that minimize unintended disruptions while ensuring successful knowledge updates.

| T-Layer | V-Layer | Rel. | T-Gen. | V-Gen. | T-Loc. | M-Loc. | Avg. |
|---|---|---|---|---|---|---|---|
| 16 | — | 97.79 | 97.23 | 97.48 | 85.28 | 71.28 | 89.81 |
| 17 | — | 97.83 | 97.41 | 97.59 | 86.71 | 71.90 | 90.29 |
| 18 | — | 97.74 | 97.26 | 97.35 | 92.40 | 71.94 | 91.34 |
| 19 | — | 97.75 | 96.75 | 97.52 | 89.18 | 80.05 | 92.25 |
| — | 16 | 95.82 | 95.35 | 95.76 | 100.00 | 76.68 | 92.72 |
| — | 17 | 96.05 | 95.31 | 95.72 | 100.00 | 76.14 | 92.64 |
| — | 18 | 95.86 | 95.30 | 95.42 | 100.00 | 75.58 | 92.43 |
| — | 19 | 95.89 | 94.59 | 95.49 | 100.00 | 77.96 | 92.79 |

Table 1: Ablations of editing different modalities at key layers. Details of metrics please see Sec. 3.1.

## 3 The Proposed DualEdit

As analyzed in Sec. 2, independently editing different modalities at their key layers is essential while maintaining an optimal balance between reliability and locality. In this section, we first introduce the base setting and evaluation metrics in Sec. 3.1, followed by an introduction to the framework and training details of DualEdit in Sec. 3.2, aiming to address the issues discussed earlier.

### 3.1 Preliminary

Consider a vision language model $f_\theta$, where $\theta$ represent the model parameters. This VLM model $f_\theta$ maps the visual inputs $\mathbf{x}_v$ and textual input $\mathbf{x}_t$ to the original output $\mathbf{o}$ (i.e., $f_\theta(\mathbf{x}_v, \mathbf{x}_t) = \mathbf{o}$ ). For a given edit sample $(\mathbf{x}_v^e, \mathbf{x}_t^e, \mathbf{o}^e)$ where $f_\theta(\mathbf{x}_v^e, \mathbf{x}_t^e) \neq \mathbf{o}^e$, the VLMs editor $M_E(\cdot)$ can generate the edited model $f_{\theta_e} = M_E(f_\theta, \mathbf{x}_v^e, \mathbf{x}_t^e, \mathbf{o}^e)$ such that $f_{\theta_e}(\mathbf{x}_v^e, \mathbf{x}_t^e) = \mathbf{o}^e$. Meanwhile, a good editor $M_E(\cdot)$ should also satisfy the following several criteria (Chen et al., 2024b; Cheng et al., 2023).

**Reliability (Rel.)** evaluates the accuracy of the post-edit model $f_{\theta_e}$ on edit samples:

$$\mathbb{E}_{(\mathbf{x}_v^e, \mathbf{x}_t^e, \mathbf{o}^e) \sim D_e} \mathbb{I}\{f_{\theta_e}(\mathbf{x}_v^e, \mathbf{x}_t^e) = \mathbf{o}^e\}, \tag{1}$$

where $D_e$ refers to the set of edit samples, and $\mathbb{I}\{\cdot\}$ is the indicator function.

**Generality (Gen.)** requires the edited model $f_{\theta_e}$ to accurately predict the correct output for inputs relevant to the exact edited samples. In the context of VLMs, generality can be further categorized into Textual Generality and Visual Generality. Textual Generality (T-Gen.) ensures that the model editing is robust to semantically equivalent variations in textual input. This reflects whether the edited model can correctly respond to the paraphrases or variations of a specific textual input. Similarly, Visual Generality (V-Gen.) ensures that the model editing remains effective across semantically equivalent variations in visual input. These can be individually expressed as:

$$
\begin{aligned}
\text{(T-Gen.)} \quad & \mathbb{E}_{(\mathbf{x}_v^e, \mathbf{x}_t^e, \mathbf{o}^e) \sim D_e} \mathbb{E}_{\hat{\mathbf{x}}_t^g \sim \mathcal{N}(\mathbf{x}_t^e)} \mathbb{I}\{f_{\theta_e}(\mathbf{x}_v^e, \hat{\mathbf{x}}_t^g) = \mathbf{o}^e\}, \\
\text{(V-Gen.)} \quad & \mathbb{E}_{(\mathbf{x}_v^e, \mathbf{x}_t^e, \mathbf{o}^e) \sim D_e} \mathbb{E}_{\hat{\mathbf{x}}_v^g \sim \mathcal{N}(\mathbf{x}_v^e)} \mathbb{I}\{f_{\theta_e}(\hat{\mathbf{x}}_v^g, \mathbf{x}_t^e) = \mathbf{o}^e\},
\end{aligned}
\tag{2}
$$

where $\mathcal{N}(\cdot)$ means the neighborhood of various edit inputs.

**Locality (Loc.)** ensures that the edited model $f_{\theta_e}$ maintains consistency with the original model $f_\theta$ for samples unrelated to the edited samples. Similar to Generality in VLMs, Locality also consists of two items, Textual Locality (T-Loc.) and Multimodal Locality (M-Loc.). The T-Loc. measures whether the edited model produces the same output as the

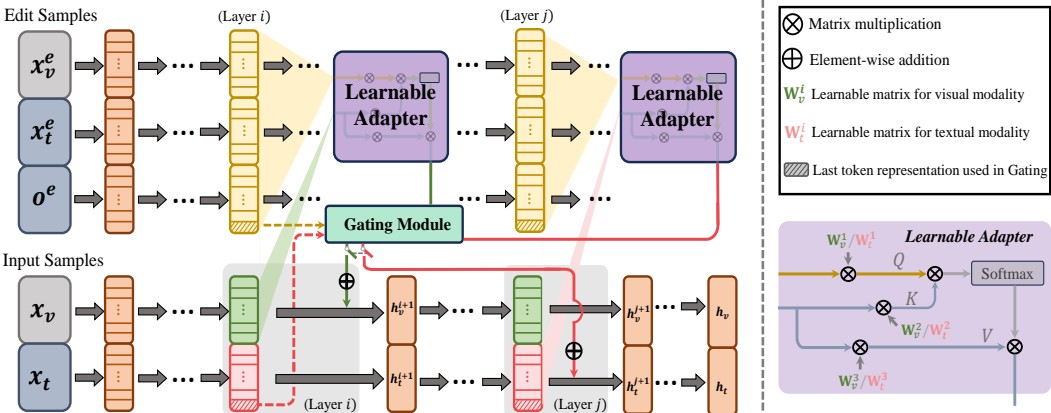

Figure 3: The framework of the proposed DualEdit method. Two modality-specific learnable adapters are inserted at designated layers. The gating module uses the representation of the last tokens to jointly control the activation of both adapters.

original model when handling text-only samples which are unrelated to the edited samples, while M-Loc. reflects whether the model retains the original outputs when both visual and textual inputs are unrelated to the edited samples, as shown in the following equations:

$$
\begin{aligned}
\text{(T-Loc.)} \quad & \mathbb{E}_{(\mathbf{x}_v^e, \mathbf{x}_t^e, \mathbf{o}^e) \sim D_e} \mathbb{E}_{(\hat{\mathbf{x}}_t^l, \hat{\mathbf{o}}^l) \sim \mathcal{U}(\mathbf{x}_t^e)} \mathbb{I}\{ f_{\theta_e}(\varnothing, \hat{\mathbf{x}}_t^l) = f_\theta(\varnothing, \hat{\mathbf{x}}_t^l) = \hat{\mathbf{o}}^l \}, \\
\text{(M-Loc.)} \quad & \mathbb{E}_{(\mathbf{x}_v^e, \mathbf{x}_t^e, \mathbf{o}^e) \sim D_e} \mathbb{E}_{(\hat{\mathbf{x}}_v^l, \hat{\mathbf{x}}_t^l, \hat{\mathbf{o}}^l) \sim \mathcal{U}(\mathbf{x}_v^e, \mathbf{x}_t^e)} \mathbb{I}\{ f_{\theta_e}(\hat{\mathbf{x}}_v^l, \hat{\mathbf{x}}_t^l) = f_\theta(\hat{\mathbf{x}}_v^l, \hat{\mathbf{x}}_t^l) = \hat{\mathbf{o}}^l \},
\end{aligned}
\tag{3}
$$

where $\mathcal{U}(\cdot)$ represent sample sets unrelated to the edited sample.

## 3.2 The Designed DualEdit Algorithm

Based on the findings discussed in Sec. 2, we propose the DualEdit algorithm, which enables editing of both modalities at their respectively important layers. To achieve a balanced trade-off between Rel and Loc. performance, we introduce a gating mechanism that leverages the cosine similarity of the last token representations. The framework is illustrated in Figure 3.

**Gating mechanism.** As shown in Figure 3, we design a gating module that determines whether to apply the editing operation. If it can effectively distinguish edit samples, the gating module function operates similarly to a Mixture of Experts (Cai et al., 2024): edit samples are processed through a dedicated learnable adapter (i.e., edited model $f_{\theta_e}$) to enhance reliability, while other samples are handled by the original model $f_\theta$, ensuring high Loc. performance.

The key question in designing a gating mechanism is how to accurately distinguish whether the input samples are edit samples. Inspired by some studies (BehnamGhader et al., 2024; Sheng et al., 2024) showing that the latent space of LLMs inherently contains rich feature information, we leverage this property in our gating mechanism by directly computing the similarity between the edit sample and the input sample in the latent space. Specifically, for a given layer, let the last token representation of the edit sample and the input sample be represented as $\mathbf{h^e}, \mathbf{h^i} \in \mathbb{R}^d$ where d is the dimension of the hidden space. Simply yet effectively, we calculate the cosine similarity of the two last token representations to determine the gating threshold by the following equation:

$$
\text{Sim} = \frac{\mathbf{h^e} \cdot \mathbf{h^i}}{\|\mathbf{h^e}\|_2 \cdot \|\mathbf{h^i}\|_2}.
\tag{4}
$$

As shown in Figure 4 (c), our gating strategy, which computes the cosine similarity of the last token representations, can effectively differentiate between editing examples (indicated by

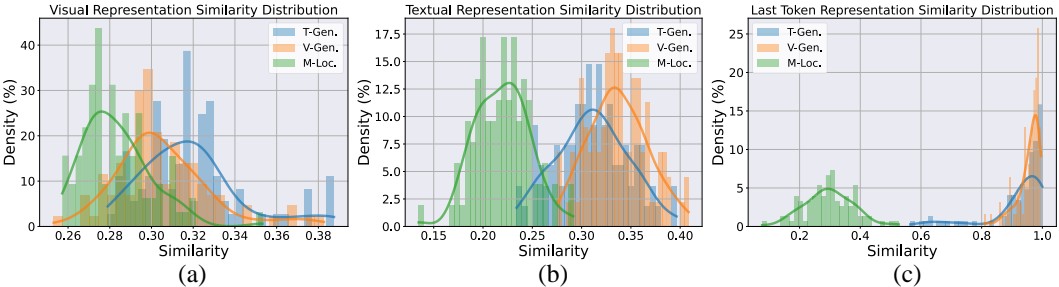

Figure 4: Gating module analysis: cosine similarity distribution of different types of representations across different samples. (a), (b) and (c) are similarity distributions calculated based on visual representations, textual representations, and the last representations.

Gen. performance) and original input samples (reflected by Loc. performance) by applying an appropriate threshold $\tau$. To further verify the effectiveness of our proposed gating module based on the last token representation, we also plotted histograms showing the similarity of regular textual and visual representations. As shown in Figure 4 (a) and (b), compared to our last-token-based approach, the regular textual and visual representations fail to effectively distinguish between the examples.

**Learnable adapters across key layers in different modalities.** By introducing the gating method, we can effectively improve Loc. performance. To further enhance the editing performance, we investigate the impact of reliability at different layers, as shown in Figure 5. The findings indicate that the 16th textual layer and the 19th visual layer deliver the best outcomes. Therefore, in our implementation, we focus on layer $i$=16 and layer $j$=19 as outlined in the flowchart of Figure 3.

For the learnable adapter, as shown in Figure 3, it includes two inputs: (1) the $k$-th layer representations $\mathbf{h}_e^k$ obtained by feeding the edit sample $(\mathbf{x}_v^e, \mathbf{x}_t^e, \mathbf{o}^e)$ into the model $f_\theta$; (2) the $k$-th layer textual representation $\mathbf{h}_t^k$ or visual representation $\mathbf{h}_v^k$. The output is the edited $k$-th layer textual or visual representation. For simplicity, we utilize cross-attention as the learnable adapter, which introduces separate learnable weights for different modalities. The concrete equation can be expressed as,

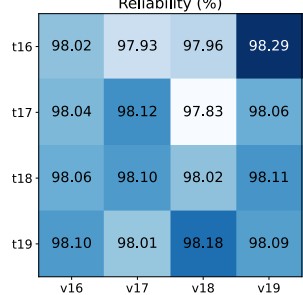

Figure 5: Visualization of the sensitivity of learnable adapters across different layers.

$$\hat{\mathbf{h}}_{t/v}^k = Softmax\left(\mathbf{h}_{t/v}^k \mathbf{W}_1^{t/v} \cdot (\mathbf{h}_e^k \mathbf{W}_2^{t/v})^\top\right) \cdot \mathbf{h}_e^k \mathbf{W}_3^{t/v},$$

where $\hat{\mathbf{h}}_t^k$ and $\hat{\mathbf{h}}_v^k$ denote the edited $k$-th layer textual and visual representations, respectively. The notation $\mathbf{W}_i^{t/v}, i \in \{1, 2, 3\}$ represent the learnable textual or visual projection matrices.

**Loss functions.** Following Cheng et al. (2023), the training loss $\ell$ of the proposed DualEdit includes three components: reliability loss $\ell_{\text{rel}}$, generality loss $\ell_{\text{gen}}$ and locality loss $\ell_{\text{loc}}$, which can be expressed as:

$$\ell = \ell_{\text{rel}} + \ell_{\text{gen}} + \ell_{\text{loc}}, \tag{5}$$

where the details of individual terms of $\ell_{\text{rel}}$, $\ell_{\text{gen}}$ and $\ell_{\text{loc}}$ are listed in Appendix A.

## 4 Experiments

### 4.1 Experimental Settings

We evaluate on E-VQA and E-IC datasets (Cheng et al., 2023; Chen et al., 2024b). For VLM backbones, we select BLIP2-OPT-2.7B (Li et al., 2023) and LLaVA-V1.5-7B (Liu et al., 2023a), which differ in architecture and training: BLIP2 uses a Q-Former and contrastive pretraining,

| Methods | E-VQA | | | | | Avg. | E-IC | | | | | Avg. |
|---|---|---|---|---|---|---|---|---|---|---|---|---|
| | Rel. | T-Gen. | V-Gen. | T-Loc. | M-Loc. | | Rel. | T-Gen. | V-Gen. | T-Loc. | M-Loc. | |
| *BLIP2-OPT* (Li et al., 2023) | | | | | | | | | | | | |
| FT-V (Cheng et al., 2023) | 24.01 | 16.00 | 20.22 | 100.00 | 88.65 | 49.78 | 42.11 | 40.74 | 36.43 | 100.00 | 89.73 | 61.80 |
| FT-L (Cheng et al., 2023) | 24.86 | 16.39 | 20.57 | 98.92 | 89.61 | 50.07 | 41.61 | 40.31 | 37.41 | 99.35 | 88.70 | 61.48 |
| KE (De Cao et al., 2021) | 67.80 | 63.00 | 66.17 | 97.32 | 45.89 | 68.04 | 69.00 | 62.80 | 61.22 | 96.21 | 45.55 | 66.96 |
| IKE (Li et al., 2023) | 99.95 | 91.59 | 92.33 | 13.16 | 1.88 | 59.78 | 96.70 | 78.20 | 83.15 | 13.36 | 2.17 | 54.72 |
| SERAC (Mitchell et al., 2022) | 91.20 | 91.40 | 89.81 | 100.00 | 0.33 | 74.55 | 94.40 | 96.00 | 91.49 | 100.00 | 0.47 | 76.47 |
| MEND (Mitchell et al., 2021) | 92.60 | 90.80 | 91.94 | 96.07 | 65.15 | 87.31 | 65.00 | 38.00 | 36.19 | 92.67 | 55.72 | 57.52 |
| TP (Huang et al., 2023) | 68.31 | 60.88 | 56.35 | 98.49 | 85.27 | 73.86 | 49.71 | 49.03 | 45.46 | 93.88 | 80.88 | 63.79 |
| LTE (Jiang et al., 2024) | 97.74 | 97.21 | 96.35 | 94.34 | 84.99 | 94.13 | 96.69 | 95.26 | 94.06 | 95.25 | 87.68 | 93.79 |
| VisEdit (Chen et al., 2024b) | 97.34 | 96.62 | 96.96 | 100.00 | 84.90 | 95.16 | 95.18 | 95.00 | 94.24 | 100.00 | 93.11 | 95.51 |
| **DualEdit (Ours)** | **98.29** | **97.89** | **98.05** | **100.00** | **99.89** | **98.82** | **96.86** | **96.72** | **96.27** | **100.00** | **100.00** | **97.97** |
| *LLaVA-V1.5* (Liu et al., 2023a) | | | | | | | | | | | | |
| FT-V (Cheng et al., 2023) | 31.68 | 29.96 | 26.68 | 100.00 | 91.23 | 55.91 | 52.85 | 51.57 | 48.63 | 100.00 | 92.55 | 69.12 |
| FT-L (Cheng et al., 2023) | 31.78 | 30.02 | 26.91 | 99.94 | 92.03 | 56.14 | 53.00 | 51.02 | 49.29 | 98.91 | 94.89 | 69.42 |
| KE (De Cao et al., 2021) | 85.86 | 84.00 | 82.23 | 93.57 | 73.06 | 83.74 | 83.54 | 82.15 | 81.12 | 92.46 | 73.83 | 82.62 |
| IKE (Li et al., 2023) | 91.35 | 90.84 | 91.08 | 60.18 | 51.08 | 76.91 | 93.72 | 88.37 | 76.99 | 76.60 | 64.90 | 80.12 |
| SERAC (Mitchell et al., 2022) | 82.51 | 81.60 | 80.05 | 100.00 | 57.48 | 80.33 | 43.08 | 42.37 | 42.85 | 100.00 | 7.63 | 47.19 |
| MEND (Mitchell et al., 2021) | 92.30 | 92.16 | 92.10 | 90.30 | 81.13 | 89.60 | 93.76 | 93.46 | 92.14 | 91.60 | 87.59 | 91.71 |
| TP (Huang et al., 2023) | 38.68 | 36.27 | 31.26 | 95.31 | 91.41 | 58.59 | 59.07 | 57.01 | 55.51 | 64.79 | 89.26 | 65.13 |
| LTE (Jiang et al., 2024) | 94.16 | 93.54 | 93.06 | 83.76 | 81.65 | 89.23 | 93.60 | 92.38 | 91.18 | 85.54 | 88.49 | 90.24 |
| VisEdit (Chen et al., 2024b) | 95.78 | 94.21 | 94.37 | 100.00 | 91.11 | 95.09 | 95.06 | 94.87 | 94.35 | 100.00 | 95.23 | 95.90 |
| **DualEdit (Ours)** | **96.94** | **96.43** | **96.20** | **100.00** | **99.61** | **97.84** | **96.76** | **96.52** | **96.24** | **100.00** | **99.74** | **97.85** |

Table 2: Main results of BLIP2-OPT and LLaVA-V1.5 on E-VQA and E-IC datasets. The best result is marked in **bold**.

while LLaVA adopts a projection layer and instruction tuning. As VisEdit is the only editing method tailored for VLMs, we compare it against adapted LLM editors including FT-V, FT-L, KE, IKE, SERAC, MEND, TP, and LTE.

## 4.2 Main Results

**Overall analysis.** According to Table 2, DualEdit demonstrates superior performance compared to other editors across both datasets. DualEdit achieves the highest average scores in both E-VQA and E-IC across different backbones, outperforming all single-modality editors. These results validate the significance of the modifications on different modalities.

**Superior M-Loc. performance.** DualEdit excels particularly in the M-Loc. metric, achieving near-perfect scores (99.89% and 100.00% with BLIP2-OPT; 99.61% and 99.74% with LLaVA-V1.5) across both datasets and backbones. The exceptional M-Loc. performance demonstrates the effectiveness of our novel gating mechanism based on the last token representation cosine similarity, which enables precise protection of locality-related samples. This approach allows VLMs to intelligently identify which content should remain unchanged while still implementing necessary edits, effectively solving a key challenge in model editing. In contrast, methods like MEND show extremely poor M-Loc. performance, indicating widespread unintended modifications that compromise model integrity.

## 4.3 Ablation Analysis

We conduct a comprehensive ablation study to evaluate the effectiveness of different components of DualEdit, with results presented in Table 3. The ablation experiments examine: (1) the impact of editing at different layers (early, middle, and late layers) in the model, (2) the effectiveness of dual-path editing compared to single-path editing, and (3) the influence of our proposed gating mechanism.

**Impact of editing at different layers.** Rows 1-5 in Table 3 show the performance when applying edits at different depth levels without the gating mechanism. The results demonstrate that editing performance varies significantly depending on which layers are targeted. When edits are applied to very early layers (T-Layer=1, V-Layer=2), average performance is limited (74.91% on E-VQA and 79.24% on E-IC). Performance improves as we move toward middle layers (T-Layer=5, V-Layer=5) and reaches its peak when targeting intermediate layers (T-Layer=10, V-Layer=10), achieving 88.58% on E-VQA and 88.25% on E-IC. Interestingly, editing very deep layers (T-Layer=30, V-Layer=30) leads to a substantial drop in performance (60.97% on E-VQA and 70.93% on E-IC). Row 6 shows our empirically

| T-Layer | V-Layer | Gating | E-VQA | | | | | Avg. | E-IC | | | | | Avg. |
|---------|---------|--------|-------|-------|-------|-------|-------|------|------|-------|-------|-------|-------|------|
| | | | Rel. | T-Gen. | V-Gen. | T-Loc. | M-Loc. | | Rel. | T-Gen. | V-Gen. | T-Loc. | M-Loc. | |
| 1 | 2 | ✗ | 93.50 | 93.34 | 92.76 | 63.65 | 31.32 | 74.91 | 79.49 | 79.50 | 76.40 | 86.63 | 74.17 | 79.24 |
| 2 | 2 | ✗ | 94.03 | 93.60 | 93.10 | 59.48 | 26.50 | 73.34 | 82.25 | 82.19 | 79.69 | 85.79 | 74.13 | 80.81 |
| 5 | 5 | ✗ | 96.72 | 96.63 | 96.35 | 65.67 | 26.92 | 76.46 | 84.70 | 84.75 | 82.57 | 82.98 | 73.58 | 81.72 |
| 10 | 10 | ✗ | 97.98 | 97.56 | 97.91 | 81.77 | 67.70 | 88.58 | 93.11 | 93.06 | 92.00 | 87.58 | 75.52 | 88.25 |
| 30 | 30 | ✗ | 54.99 | 52.40 | 52.26 | 88.88 | 56.31 | 60.97 | 57.48 | 56.27 | 52.83 | 95.19 | 92.89 | 70.93 |
| 16 | 19 | ✗ | 98.29 | 97.89 | 98.05 | 90.53 | 72.02 | 91.36 | 96.86 | 96.72 | 96.27 | 100.00 | 92.05 | 96.38 |
| 16 | — | ✓ | 97.79 | 97.23 | 97.48 | 100.00 | 99.88 | 98.48 | 82.61 | 82.41 | 80.49 | 99.35 | 98.85 | 88.74 |
| — | 19 | ✓ | 95.89 | 94.59 | 95.49 | 100.00 | **99.90** | 97.17 | 94.00 | 93.96 | 93.19 | 100.00 | 100.00 | 96.23 |
| 16 | 19 | ✓ | **98.29** | **97.89** | **98.05** | **100.00** | 99.89 | **98.82** | **96.86** | **96.72** | **96.27** | **100.00** | **100.00** | **97.97** |

Table 3: Ablation results of BLIP2-OPT on E-VQA and E-IC datasets, exploring different layers and the impact of our proposed gating module. The best result is marked in **bold**.

determined optimal layer configuration (T-Layer=16, V-Layer=19) without gating, which achieves strong performance of 91.36% on E-VQA and 96.38% on E-IC.

**Dual-Editing vs. Single-Editing.** Rows 7-8 examine the effectiveness of our dual-editing approach compared to single-editing alternatives, both with the gating mechanism enabled. In row 7, we apply edits only to the textual representations, while in row 8, we apply edits only to the visual representations. Our full DualEdit approach (row 9), which simultaneously edits both textual and visual representations (T-Layer=16, V-Layer=19) with gating, consistently outperforms single-editing approaches across both datasets, achieving 98.82% on E-VQA and 97.97% on E-IC. This demonstrates the complementary benefits of modifying both modality paths, allowing the model to integrate edited knowledge more comprehensively.

**Effectiveness of gating mechanism.** The most significant improvement comes from our proposed gating mechanism, as evidenced by comparing rows 6 and 9, which have identical layer configurations (T-Layer=16, V-Layer=19) but differ in whether gating is enabled. Without gating (row 6), the model achieves 91.36% on E-VQA and 81.77% on E-IC. With gating (row 9), performance jumps significantly to 98.82% on E-VQA (+7.46%) and 97.97% on E-IC (+16.20%). The benefits of gating are particularly evident in the locality metrics (T-Loc. and M-Loc.), which measure the model's ability to preserve unrelated knowledge. Without gating, M-Loc. scores are 72.02% on E-VQA and 92.05% on E-IC. With gating enabled, these scores improve dramatically to 99.89% and 100.00%, respectively, approaching perfect preservation of unrelated knowledge. This substantial improvement confirms that our gating mechanism successfully prevents unwanted modifications to non-target knowledge while allowing precise edits on target knowledge.

Overall, our ablation study highlights three key insights. First, layer selection is critical: editing intermediate layers yields the best performance, while very early or very deep layers lead to suboptimal results. Second, modality-specific edits are complementary: editing only one modality benefits certain tasks, but dual-editing consistently achieves the best overall performance. Third, and most notably, our gating mechanism plays a crucial role in balancing knowledge integration with preservation.

# 5 Related Works

## 5.1 Vision Languages Model

Earlier vision-language models, exemplified by CLIP (Radford et al., 2021), establish alignment between textual and visual information within shared hidden spaces through contrastive learning on extensive datasets. These models demonstrate remarkable generalization across diverse tasks with minimal adaptation requirements. Building upon these foundations, VLMs have successfully bridged the gap between visual and linguistic modalities, achieving exceptional results in various applications including in-context predictions (Liu et al., 2023b; Salewski et al., 2023), multi-image understanding, and chain-of-thought reasoning (Driess et al., 2023; Yang et al., 2023). The landscape of Large VLMs encompasses diverse architectural designs that reflect varying approaches to multimodal integration and

processing (Wadekar et al., 2024). Token Fusion represents an architectural paradigm where tokenized input modalities are fed directly into the model's input stage. This approach employs either a decoder-only transformer or an encoder-decoder style transformer as the multimodal integration mechanism, as exemplified by models like LaVIT (Jin et al., 2024). Deep Fusion approaches incorporate visual information into the internal layers of the LLM through cross-attention mechanisms. While models like PaLI-X (Chen et al., 2023) and Flamingo (Alayrac et al., 2022) implement standard cross-attention layers, alternatives such as LLaMA-Adapter(Zhang et al., 2024b) utilize custom-designed components to process visual representations before cross-attention operations.

Early Projection Fusion represents another prevalent strategy where non-tokenized visual inputs undergo processing before being introduced at the model's input rather than within internal layers. Various connection modules facilitate this integration, including linear projection layers, Q-formers with linear projections, perceiver resamplers, and custom learnable components. Notable implementations of this approach include Qwen2.5-VL (Bai et al., 2025), LlavaV1.5 (Liu et al., 2024), and MiniGPT4 (Zhu et al., 2023a), which have demonstrated superior capabilities in multimodal understanding and generation. This paper primarily focuses on examining and enhancing editing capabilities within this class of vision-language models.

## 5.2 Model Editing

Model editing in large language models (LLMs) lies at the intersection of *continual learning* (Wu et al., 2024a;b; 2025) and *parameter-efficient fine-tuning* (Si et al., 2025), aiming to incorporate new factual knowledge or behavioral changes into models with minimal forgetting and computational cost. From the CL perspective, model editing seeks to update model knowledge while mitigating catastrophic forgetting (Lopez-Paz & Ranzato, 2017), whereas from the PEFT angle, it emphasizes modifying only a small subset of parameters to achieve targeted updates (Hu et al., 2022). Recent advances in model editing for LLMs can be broadly classified into three paradigms: *parameter modification*, *module-based augmentation*, and *prefix-based instruction injection*. *Parameter modification* methods aim to directly adjust the internal weights of a model in response to specific edit instructions. Among these, Knowledge Editor (KE) (De Cao et al., 2021) and MEND (Mitchell et al., 2021) adopt a learning-based approach, where an auxiliary network is trained to generate weight deltas based on edit signals. On the other hand, ROME (Meng et al., 2022a) and MEMIT extension (Meng et al., 2022b) leverage tools from causal inference. A different line of work explores *module-based augmentation* to achieve editing without overwriting existing model knowledge. For instance, SERAC (Mitchell et al., 2022) trains an edit-aware counterfactual model that only activates under relevant conditions. TP (Huang et al., 2023) introduces an editable "knowledge neuron" that can be trained separately from the base model. GRACE (Hartvigsen et al., 2023) routes inputs to a target editing output in latent space, conditional on their similarity crossing a predefined threshold. MELO (Yu et al., 2024) builds upon GRACE by retrieving and injecting editing matrices related to the input query, thereby enabling efficient updates to the model's predictions. In contrast, *prefix-tuning approaches* avoid changing model weights by manipulating the context seen during inference. IKE (Zheng et al., 2023) applies in-context learning to guide the model's output based on a few-shot edited prompt, while LTE (Jiang et al., 2024) explicitly trains the model to follow editing instructions. RECIPE (Chen et al., 2024a) further advances this line by introducing a learnable prompt generator that finds the shortest continuous prefix capable of inducing the desired model behavior.

Although VLMs have gained increasing attention, research on editing such models remains relatively underexplored (Cheng et al., 2023). VisEdit (Chen et al., 2024b) represents the first attempt to identify and edit key layers within the visual modality. However, different from our DualEdit, it concentrates solely on the visual modality and completely neglects the textual modality, which fails to recognize the influence of different modalities in VLMs.

## 6 Conclusion

In this work, we investigate the largely unexplored problem of editing VLMs and highlight the distinct roles that textual and visual modalities play in this process. Through comprehensive empirical analysis, we reveal that textual and visual representations exhibit different sensitivity patterns across layers, and we also discover that editing both modalities can effectively update the model's knowledge, but it may compromise its original capabilities. Motivated by these findings, we propose DualEdit, a modality-aware editing framework that applies targeted updates to each modality at its respective key layer. Additionally, a gating mechanism based on the similarity of the last token representation helps DualEdit achieve a better balance between knowledge incorporation (i.e., reliability and generality) and preservation (i.e., locality). Extensive experiments across multiple VLM backbones and datasets demonstrate that DualEdit consistently outperforms both VLM-specific and adapted LLM editing baselines, setting a new standard for multimodal model editing.

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

# A  Loss Function

The individual loss functions $\ell_{\text{rel}}$, $\ell_{\text{gen}}$ and $\ell_{\text{loc}}$ are expressed as,

$$\ell_{\text{rel}} = \sum_{(\mathbf{x}_v^e, \mathbf{x}_t^e, \mathbf{o}^e) \sim D_e} - \log f_{\theta_e}(\mathbf{o}^e | \mathbf{x}_v^e, \mathbf{x}_t^e),$$

$$\ell_{\text{gen}} = \sum_{\substack{(\mathbf{x}_v^e, \mathbf{x}_t^e, \mathbf{o}^e) \sim D_e \\ \hat{\mathbf{x}}_t^g, \hat{\mathbf{x}}_v^g \sim \mathcal{N}(\mathbf{x}_t^e, \mathbf{x}_v^e)}} - \log f_{\theta_e}(\mathbf{o}^e | \hat{\mathbf{x}}_v^g, \mathbf{x}_t^e) - \log f_{\theta}(\mathbf{o}^e | \mathbf{x}_v^e, \hat{\mathbf{x}}_t^g),$$

$$\ell_{\text{loc}} = \sum_{\substack{(\mathbf{x}_v^e, \mathbf{x}_t^e, \mathbf{o}^e) \sim D_e \\ (\hat{\mathbf{x}}_v^l, \hat{\mathbf{x}}_t^l, \hat{\mathbf{o}}^l) \sim \mathcal{U}(\mathbf{x}_v^e, \mathbf{x}_t^e)}} \text{KL}\left(f_{\theta}\left(\hat{\mathbf{o}}^l | \hat{\mathbf{x}}_v^l, \hat{\mathbf{x}}_t^l\right) || f_{\theta_e}\left(\hat{\mathbf{o}}^l | \hat{\mathbf{x}}_v^l, \hat{\mathbf{x}}_t^l\right)\right) + \text{KL}\left(f_{\theta}\left(\hat{\mathbf{o}}^l | \varnothing, \hat{\mathbf{x}}_t^l\right) || f_{\theta_e}\left(\hat{\mathbf{o}}^l | \varnothing, \hat{\mathbf{x}}_t^l\right)\right).$$

# B  Dataset Information

E-VQA focuses on editing visual question answering capabilities, where the dataset contains examples requiring models to correct their answers to questions about images. The training set includes 6,346 examples, with 2,093 in the test set. E-IC targets image captioning abilities, containing examples where models need to improve or correct their image descriptions. It has 2,849 examples for training and 1,000 for testing.

Both datasets are evaluated on reliability (successful editing of target examples), locality (preserving unrelated knowledge), and generality (maintaining consistency with similar inputs). They're used to assess how effectively different editing methods can correct mistakes in multimodal LLMs without disrupting their overall performance.

# C  Experimental Settings

**Datasets:** Following (Chen et al., 2024b), we utilize E-VQA (Editing Visual Question Answering) and E-IC (Editing Image Caption) proposed by (Cheng et al., 2023) as our evaluation datasets.

**VLM Backbones:** BLIP2-OPT-2.7B (Li et al., 2023) employs a Q-Former architecture to align image-to-visual representations, while LLaVA-V1.5-7B (Liu et al., 2023a) uses a projection layer for alignment. BLIP2-OPT-2.7B features fewer parameters and was pre-trained with image-text contrastive learning, whereas LLaVA-V1.5-7B contains more parameters and underwent instruction tuning with conversation data. Considering these differences in architecture, parameter size, and training methodology, we selected these two models to serve as fair representatives for VLM backbones in our editor evaluation.

**Baselines:** To our knowledge, VisEdit currently stands as the only editing approach specifically designed for vision-language models. Consequently, most prior research has adapted general language model editing methods to the vision-language domain. These adapted approaches include FT-V (Fine-tunes visual encoder), FT-L (Fine-tunes last layer of the language model), KE (De Cao et al., 2021), IKE (Zheng et al., 2023), SERAC (Mitchell et al., 2022), MEND (Mitchell et al., 2021), TP (Huang et al., 2023) and LTE (Jiang et al., 2024). We utilize both vision-language model approaches and adapted large language model techniques as our comparative baselines.

# D  Implementation Details

**Choice of gating threshold:** We set the threshold $\tau$ to 0.6 when using LLaVA-V1.5 as the backbone model and 0.8 when using BLIP2-OPT as the backbone.

**Training hyper-parameters:** The optimal DualEdit insertion layers and training hyperparameters remain consistent across both BLIP-OPT and LLaVA-V1.5 models. The DualEdit module is optimally inserted at layers 16 and 19. Layer 16 is responsible for modifying the

textual representations, while layer 19 handles the modification of visual representations. We set the learning rate to 0.0001, the training batch size to 4, and the maximum number of iterations to 50000. A checkpoint is saved every 1000 iterations. We use the checkpoint with the smallest loss for evaluations. The training process requires approximately 1 day on 2 NVIDIA H100 GPUs.

