# OpenReview forum: "DualEdit: Dual Editing for Knowledge Updating in Vision-Language Models"
_colmweb.org/COLM/2025/Conference — COLM 2025_

### Official Review · Reviewer_CGGe · 2025-05-11

**Rating:** 6
**Confidence:** 4
**Ethics Flag:** 1

**Summary:**

The paper proposes DualEdit, a dual-modality editing framework for Vision-Language Models (VLMs) that addresses the underexplored challenge of updating knowledge in multimodal settings. Existing editing methods primarily focus on single-modal LLMs, but VLMs involve complex interactions between visual and textual modalities. Through empirical analysis, the authors identify that textual and visual representations exhibit varying sensitivity across different layers, with textual modalities generally more influential. DualEdit introduces modality-specific edits at key layers (e.g., textual layers at 16, visual layers at 19) and incorporates a gating mechanism based on cosine similarity of last-token representations to balance reliability (successful edits) and locality (preserving unrelated knowledge). Experiments on E-VQA and E-IC datasets with BLIP2-OPT and LLaVA-V1.5 backbones demonstrate DualEdit’s superiority over VLM-specific and adapted LLM editing baselines, achieving near-perfect locality preservation and high reliability.

**Reasons To Accept:**

- First comprehensive study decoupling modality-specific impacts in VLM editing, revealing critical insights (e.g., layer-wise sensitivity differences).

- DualEdit’s modality-aware design and gating mechanism offer a principled solution for multimodal editing, outperforming state-of-the-art methods.

-  Extensive experiments across multiple backbones (BLIP2, LLaVA) and datasets (E-VQA, E-IC) validate effectiveness, with detailed ablation studies justifying design choices.

- The authors provides the various analysis of model editing data and model modules to facilitates the readers understanding.

**Reasons To Reject:**

- Although the authors provide empirical evidence for the effectiveness of DualEdit, the theoretical foundations of the method are relatively weak. For instance, the selection of key layers for editing and the design of the gating mechanism lack rigorous theoretical analysis, making it difficult to fully understand the underlying principles of the method.

- Experiments are restricted to two datasets and backbones; broader validation (e.g., on more VLMs like Flamingo or PaLI) would strengthen claims. Also, the data analysis of Sec. 2 only uses the LLaVA-V1.5 as the backbone to conduct the VLM's layer and metrics' charateristics.

- The results of baselines are different from the original baselines' paper. For example, the results reported by VisEdit [1] are better than the metrics of Table 2.

[1] Attribution Analysis Meets Model Editing: Advancing Knowledge Correction in Vision Language Models with VisEdit

---

> ### Author Response · Authors · 2025-06-03
> **To Reviewer CGGe**
>
> We sincerely thank the reviewer for providing valuable feedback. Please kindly let us know whether you have any further questions.
>
> **Q1:** Although the authors provide empirical evidence for the effectiveness of DualEdit, the theoretical foundations of the method are relatively weak. For instance, the selection of key layers for editing and the design of the gating mechanism lack rigorous theoretical analysis, making it difficult to fully understand the underlying principles of the method.
> > We appreciate your recognition of our empirical evidence for DualEdit's effectiveness. While we agree that deeper theoretical analysis would be valuable, we would like to emphasize that understanding the internal representation mechanisms of LLMs and VLMs remains one of the most challenging open problems in the field. In light of this, our work focuses on making comprehensive and rigorous empirical contributions:
> > - **Systematic empirical discovery**: We are the first to conduct comprehensive layer-wise sensitivity analysis across both modalities during the VLM editing process, revealing that textual and visual representations reach peak sensitivity at different layers (Finding-1).
> > - **Evidence-driven design**: Our key layer selection and gating mechanism design are grounded in extensive empirical analysis (Figures 1-2, Table 1), following established practices in model editing research.
> > - **Practical effectiveness**: Our method achieves superior performance across multiple metrics and backbones, demonstrating that empirical insights can lead to effective solutions even when theoretical understanding is limited.
> >
> > We agree that building rigorous theoretical foundations for VLM editing is an important and ambitious direction for future work. Meanwhile, we believe our empirical findings offer meaningful insights that push the field forward in a concrete and actionable way.
>
>
> **Q2:** Experiments are restricted to two datasets and backbones; broader validation (e.g., on more VLMs like Flamingo or PaLI) would strengthen claims. Also, the data analysis of Sec. 2 only uses the LLaVA-V1.5 as the backbone to conduct the VLM's layer and metrics' charateristics.
> > We would like to clarify our experimental scope and architectural focus:
> > - **Fousing on specific VLM family**：As discussed in Section 5.1, our work targets **Early Projection Fusion VLMs**, where visual features are projected and fused into the language model at the input level (rather than injected into internal layers). This design is widely adopted in state-of-the-art models such as LLaVA-V1.5, BLIP2-OPT, which have demonstrated strong performance in both academic and industrial settings.
> > - **Different challenges for Deep Fusion architectures**： Extending DualEdit to Deep Fusion architectures (e.g., Flamingo, PaLI-X), **which integrate visual inputs via cross-attention into internal LLM layers, presents fundamentally different challenges**. Adapting our layer selection strategy and gating mechanism to such models would require substantial redesign, and is thus **beyond the scope of this paper**.
> > - **Additional validation in progress**: To better address the reviewer’s suggestion for broader validation, we have initiated experiments on MiniGPT-4, another representative Early Projection VLM. Due to current hardware constraints and time limitations, the experiments are still in progress.
> > - **Analysis using more backbones**: While our original analysis was based on LLaVA-V1.5, to ensure generality, we additionally analyze attention scores across different layers for BLIP2-OPT. The results are available at the following anonymous link: https://imgur.com/Kw2HSSC. According to the figure, although the attention score patterns of BLIP2-OPT differ significantly from those of LLaVA-V1.5, both are consistent with our Finding 1: Textual and visual representations in VLMs demonstrate varying levels of importance within the same layer, and the significance of each modality shifts across different layers.
>
> **Q3:** The results of baselines are different from the original baselines' paper. For example, the results reported by VisEdit [1] are better than the metrics of Table 2.
> > Thanks for your careful review. To ensure fair and accurate comparison,
> >  - We **faithfully reproduced the VisEdit results using the official implementation**, strictly following their released hyperparameters and experimental settings.
> >  - All methods, including VisEdit and our DualEdit, were **evaluated in a unified experimental environment, using the same hardware, software versions, and evaluation protocols.**
> >
> >  While our reproduced VisEdit results may differ slightly from those reported in their original paper, **the relative performance comparisons remain valid and fair**, since all methods were run under identical conditions in our study.
>
> [1] Attribution Analysis Meets Model Editing: Advancing Knowledge Correction in Vision Language Models with VisEdit

---

> > ### Comment · Area_Chair_JQmv · 2025-06-05
> > **Response needed from CGGe**
> >
> > Reviewer CGGe, please take time to consider the author's response and update your review of the paper.

---

> > > ### Comment · Reviewer_CGGe · 2025-06-06
> > >
> > > Thanks for authors' rebuttal content. After carefully reviewing the answers of weaknesses, I will keep my score.

---

> > > > ### Author Response · Authors · 2025-06-07
> > > > **We Appreciate Your Feedback**
> > > >
> > > > Thank you for reviewing our rebuttal. We appreciate your evaluation and your decision to maintain a positive score.

---

### Official Review · Reviewer_DZuA · 2025-05-12

**Rating:** 6
**Confidence:** 3
**Ethics Flag:** 1

**Summary:**

This paper addresses the challenge of model editing in Vision-Language Models (VLMs), a task largely under-explored compared to single-modal LLM editing. The authors conduct a empirical analysis to understand how visual and textual modalities contribute differently to knowledge representation and sensitivity across model layers. Based on their findings, they propose DualEdit, a novel editing framework that independently edits visual and textual features at their respective sensitive layers. A key component is a gating mechanism based on the cosine similarity of last-token representations, which helps to preserve locality while allowing targeted edits. The method is evaluated on two VLM backbones (BLIP2 and LLaVA) and two benchmarks (E-VQA and E-IC), outperforming existing baselines in reliability, generality, and locality.

**Questions To Authors:**

- Given the limited difference in attention maps across V-Layers in Table 1, is the visual editing branch necessary or could simpler alternatives suffice?
- Could the gating mechanism be adapted to perform soft gating (e.g., attention-weighted blending) rather than binary on/off activation?
- What is the sensitivity of the method to threshold selection and layer choice across different VLMs?

**Reasons To Accept:**

- The paper provides a much-needed exploration of editing in multimodal models, an area that lags behind LLM editing research.
- DualEdit achieves excellent performance on both accuracy and locality metrics, particularly through its gating mechanism.
- The evaluation includes diverse datasets, lending credibility to the results.

**Reasons To Reject:**

- Visual attention maps across layers show limited variation, raising questions about the necessity of editing at different V-Layers.
- The conceptual novelty may feel limited, as the work extends existing ideas (like adapter-based editing and modality-specific routing) rather than introducing a fundamentally new editing paradigm.

---

> ### Author Response · Authors · 2025-06-03
> **To Reviewer DZuA Q1 & Q2**
>
> We greatly appreciate your constructive comments on our paper. We hope that our response satisfactorily addresses the issues you raised.
>
> **Q1:** Given the limited difference in attention maps across V-Layers in Table 1, is the visual editing branch necessary or could simpler alternatives suffice?
> > Thank you for your careful review and for pointing this out. The limited differences in attention maps across V-Layers in Table 1 are due to our preliminary analysis, which indicated that the most critical layers are between layers 16 and 19. As a result, all these layers perform relatively well, leading to only small differences. We would like to clarify that the visual editing branch is indeed necessary, as demonstrated by the following points.
> > - **(Visual editing is necessary).** As shown in the ablation results in Table 3 (also listed below for convenience), removing the visual editing component leads to a clear drop in performance, where the average score decreases by 9.23\%. This result highlights the importance of the visual editing branch in improving model performance.
>
> | T-Layer | V-Layer | Gating | Rel. (%) | T-Gen.(%)| V-Gen.(%)| T-Loc.(%)| M-Loc.(%)| Avg.(%)|
> |---------|---------|--------|---------------|---------------|---------------|---------------|---------------|-------------|
> | 16      | —       | ✓      | 82.61         | 82.41         | 80.49         | 99.35         | 98.85         | 88.74       |
> | **16** | **19** | **✓** | **96.86** | **96.72** | **96.27** | **100.00** | **100.00** | **97.97** |
>
> > - **(The location of visual editing matters).** To further analyze the impact of the visual modality, we also provide results for inserting the visual editing branch at layers 2, 5, 10, and 30.  These results show that the choice of layer significantly affects performance, reinforcing the necessity of careful design when integrating visual information.
>
> | T-Layer | V-Layer | Gating| Rel.(%) | T-Gen.(%) | V-Gen.(%) | T-Loc.(%) | M-Loc.(%) | Avg.(%) |
> |---------|---------|--------|------------|--------|--------|--------|--------|------|
> | 16       | 2      | ✓      |84.12| 83.92  | 84.17  | 100.00 | 100.00 | 90.44|
> | 16       | 5      | ✓      |85.89| 84.96  | 84.59  | 100.00 | 100.00 | 91.09|
> | 16       | 10      | ✓     |93.21| 93.11  | 93.19  | 100.00 | 100.00 | 95.90|
> | 16       | 30      | ✓     |83.09| 82.86  | 83.01  | 100.00 | 100.00 | 89.79|
> | 16  | 19  | ✓  | **96.86** | **96.72** | **96.27** | **100.00** | **100.00** | **97.97** |
>
>
> **Q2:** Could the gating mechanism be adapted to perform soft gating (e.g., attention-weighted blending) rather than binary on/off activation?
>
> > We appreciate your interest in the concept of soft gating, which indeed inspired us to further investigate its potential.
> >
> > Following your suggestion, we implemented a soft gating mechanism where the last token of the edit sample attends to the last token of the input sample via a learnable cross-attention module. The resulting attention score is used as a continuous weight to softly combine representations, thereby enabling a dynamic gating process.
>
> | Ablation          | E-VQA Rel. | E-VQA T-Gen. | E-VQA V-Gen. | E-VQA T-Loc. | E-VQA M-Loc. | E-VQA Avg. | E-IC Rel. | E-IC T-Gen. | E-IC V-Gen. | E-IC T-Loc. | E-IC M-Loc. | E-IC Avg. |
> |---------|------------|--------|--------|--------|--------|------|-----------|--------|--------|--------|--------|------|
> | No Gating | 98.29      | 97.89  | 98.05  | 90.53  | 72.02  | 91.36| 96.86     | 96.72  | 96.27  | 100.00 | 92.05  | 96.38|
> | Soft Gating | 96.27      | 95.51  | 95.69  | 92.13 | 87.42| 93.41| 94.12     | 93.79  | 93.35  | 100.00 | 94.18 | 95.09|
> | **DualEdit (Ours)** |**98.29**  | **97.89**| **98.05**| **100.00**| **99.89**| **98.82**| **96.86** | **96.72** | **96.27** | **100.00** | **100.00** | **97.97** |
>
> > As shown in the results, soft gating indeed improves locality, demonstrating its effectiveness in adapting to edit-specific context. However, we also observe that it leads to a noticeable decrease in both reliability and generality. Compared to soft gating, our method achieves better performance. We attribute this to the following insights:
> > - **(Stable threshold selection)**. While our gating method relies on a threshold as a hyperparameter, Figure 4 shows that gating based on last-token similarity achieves a clear separation, making the threshold straightforward and stable to select. Moreover, this clear distinction helps explain why our gating mechanism outperforms soft gating.
> > - **(Inference-time only gating)**. Our gating mechanism is applied exclusively during inference, leaving the training process unaffected. This allows the model to preserve generalizable representations during learning, which in turn improves both reliability and generality.
>
> > In summary, while soft gating is an interesting direction and indeed offers certain benefits, we find that our proposed approach achieves a more favorable solution.

---

> ### Author Response · Authors · 2025-06-03
> **To Reviewer DZuA Q3**
>
> **Q3:** What is the sensitivity of the method to threshold selection and layer choice across different VLMs?
>
> > Thank you for your question regarding the sensitivity of our method to threshold selection and layer choice across different VLMs.
> > - **(Threshold sensitivity)**. We present results under different threshold values. These results demonstrate that the model is not sensitive to the threshold choice; as long as it remains below a certain critical point, both reliability and generality remain stable. This indicates that threshold setting within our proposed DualEdit method is not a significant concern.
>
> | Threshold          | E-VQA Rel. | E-VQA T-Gen. | E-VQA V-Gen. | E-VQA T-Loc. | E-VQA M-Loc. | E-VQA Avg. | E-IC Rel. | E-IC T-Gen. | E-IC V-Gen. | E-IC T-Loc. | E-IC M-Loc. | E-IC Avg. |
> |---------|------------|--------|--------|--------|--------|------|-----------|--------|--------|--------|--------|------|
> | 0.0 | 98.29      | 97.89  | 98.05  | 90.53  | 72.02  | 91.36| 96.86     | 96.72  | 96.27  | 100.00 | 92.05  | 96.38|
> | 0.3 | 98.29      | 97.89  | 98.05  | 96.26  | 86.71  | 95.44| 96.86     | 96.72  | 96.27  | 100.00 | 94.40  | 96.85|
> | 0.5 | 98.29      | 97.89  | 98.05  | 97.59  | 92.86  | 96.94| 96.86     | 96.72  | 96.27  | 100.00 | 98.45  | 97.66|
> | 0.7 | 98.29      | 97.89  | 98.05  |100.00  | 99.53  | 98.75| 96.86     | 96.72  | 96.27  | 100.00 | 100.00  | 97.97|
> | 0.9 | 82.62      | 82.51  | 82.14  | 100.00 | **100.00** | 89.94| 78.34     | 78.02  | 78.37  | 100.00 | 100.00 | 86.95|
> | **0.8 (ours)** |**98.29**  | **97.89**| **98.05**| **100.00**| 99.89| **98.82**| **96.86** | **96.72** | **96.27** | **100.00** | **100.00** | **97.97** |
>
> > - **(Layer choice sensitivity)**. As discussed in the response to Q1, we analyzed the impact of applying editing at different layers (e.g., layers 2, 5, 10, 30), and found that performance is relatively stable as long as editing is performed within the top visual layers (e.g., layers 16–19), which we identified through attention map analysis. When editing is applied to very shallow or very deep layers, performance may degrade, particularly for tasks that require stronger visual grounding (e.g., E-IC).
> >
> >  We will include this analysis in the revised version and clarify these points accordingly.

---

> > ### Comment · Area_Chair_JQmv · 2025-06-05
> > **Response needed from DZuA**
> >
> > Reviewer DZuA, please take time to consider the author's response and update your review of the paper.

---

> ### Author Response · Authors · 2025-06-09
> **Looking Forward to Receiving Feedback from Reviewer DZuA**
>
> Thank you again for your thoughtful review. We would like to follow up to see if our responses have addressed your concerns or if you have any further questions. We would really appreciate the opportunity to discuss this further if our responses have not yet addressed your concerns.

---

> > ### Comment · Reviewer_DZuA · 2025-06-09
> >
> > I have read all your response and thanks for your efforts. I would keep my positive score.

---

> > > ### Author Response · Authors · 2025-06-09
> > > **We Appreciate Your Feedback**
> > >
> > > Thank you for reviewing our rebuttal. We appreciate your evaluation and your decision to maintain a positive score.

---

### Official Review · Reviewer_Ykrx · 2025-05-13

**Rating:** 6
**Confidence:** 4
**Ethics Flag:** 1

**Summary:**

Most existing model editing methods are designed for models that only have one modality. These methods cannot be applied to VLM efficiently. In order to refine the VLM behaviors, this paper made an assumption that each modality in VLM plays a different role in knowledge editing. The analysis in this paper focuses on two perspectives: 1. Attention scores of different modalities at various layers; 2. How to maintain the reliability when editing specific modalities. Based these two findings, this paper proposes DualEdit method that considers both textual and visual modalities. The key is to propose a gating module using a learnable adapter.

**Questions To Authors:**

Please answer the questions listed in "reasons to reject".

**Reasons To Accept:**

- This paper proposed a comprehensive analysis on the importances of various modalities in VLM. It provides some insights about identifying the role of modalities cross layers and within layers.

- The proposed method DualEdit ensures a balanced tradeoff between reliability and locality.

- This proposed method shows very reasonable performance across various VLM benchmarking datasets

**Reasons To Reject:**

- These two findings have been explored in some papers before. For example, there are many papers that want to emphasize the visual modality because they found the VLMs haven’t fully captured the image information. In addition, balancing the reliability and locality is an important aspect in the model editing methods. Hence these two findings have been proposed before. The novelty of this analysis is limited.

- Training the adapters needs the data. Preparing the pairs including original sample and edit sample will be a time-consuming task. This paper hasn’t explained how to prepare these edit samples clearly. This is one of the most important part this paper needs to clarify. It is better to explain if we can prepare the data for model training easily.

- If we trained an adapter using one type of data, I am not sure if it can be adapted a domain that is totally difference with the training data. Providing the generalization analysis of proposed method is needed for this paper

---

> ### Author Response · Authors · 2025-06-03
> **To Reviewer Ykrx Q1 & Q2**
>
> Thanks for your time in dealing with our work. We will answer the question and discuss point by point as follows. We hope that our response satisfactorily addresses the issues you raised. Please feel free to let us know if you have any additional concerns or questions.
>
> **Q1**: These two findings have been explored in some papers before. For example, there are many papers that want to emphasize the visual modality because they found the VLMs haven’t fully captured the image information. In addition, balancing the reliability and locality is an important aspect in the model editing methods. Hence these two findings have been proposed before. The novelty of this analysis is limited.
> > We would like to clarify that there is a basic misunderstanding about the two findings.
> > - **(The first finding)**. To the best of our knowledge, our work is the first to analyze the influence of both textual and visual modalities simultaneously during the editing process. Unlike previous works that only emphasize the visual modality, we systematically investigate how sensitivity to edits varies across layers for both modalities, and identify modality-specific editing layers.
> > - **(The second finding)**. We separately study how each modality affects locality, reliability, and generality, with the goal of understanding the distinct roles of different modalities. This is different from prior work, which mainly emphasizes the importance of balancing reliability and locality, without explicitly analyzing the contribution of each modality.
> > - Our work does not merely reiterate the insights outlined in Q1 but extends them by introducing modality-aware, layer-sensitive editing strategies specifically tailored for VLMs, which we believe represents a meaningful advancement in the field.
> >
> > Building on the two findings, we propose DualEdit, a novel editor that explicitly modifies both modalities at their respective critical layers. DualEdit incorporates a gating mechanism to preserve the model’s original capabilities, particularly addressing the reliability–locality trade-off in the context of dual-modality editing
>
> **Q2**: Training the adapters needs the data. Preparing the pairs including original sample and edit sample will be a time-consuming task. This paper hasn’t explained how to prepare these edit samples clearly. This is one of the most important part this paper needs to clarify. It is better to explain if we can prepare the data for model training easily.
> > Thank you for pointing this out. We would like to further clarify the details of the edit samples and benchmarks.
> > - In line with prior VLM model editing works [1-2], we directly use their publicly available benchmarks which contains pre-constructed original-edit sample pairs.
> > - We do not introduce any extra paired data for the training of our proposed adapter. Therefore, the data preparation process is not more time-consuming than that of previous works.
> > - We will revise the corresponding section of the paper to improve the clarity of our data preparation process and make this point more explicit.
>
> [1] Can we edit multimodal large language models?
>
> [2] Attribution analysis meets model editing: Advancing knowledge correction in vision language models with visedit

---

> ### Author Response · Authors · 2025-06-03
> **To Reviewer Ykrx Q3**
>
> **Q3**: If we trained an adapter using one type of data, I am not sure if it can be adapted a domain that is totally difference with the training data. Providing the generalization analysis of proposed method is needed for this paper.
> > Thanks for the comment.  To better illustrate the generalizability of the proposed DualEdit, we would like to provide clarification from both in-domain and out-of-domain perspectives:
> > -  Let $\\theta$ denote the parameters of the traned model $f(\\cdot)$. During training, the model is updated based on a set of editing samples $x$, resulting in new parameters $\\theta'$, such that the model can correct specific outputs while preserving other outcomes. During testing, we hope the $\\theta'$ can adapt to new editing samples $x'$, enabling the model to edit them correctly without affecting the outputs for other inputs.
> >
> > - (**In-Domain Generalizability**). [**The domains of $x'$ and  $x$ are similar.**] As shown in Table 2 for E-VQA and E-IC, our DualEdit achieves the best performance on their respective test datasets. For example, our DualEdit ourperforms VisEdit by 3.8 in average on E-VQA with BLIP backbone. This clearly demonstrates superior in-domain generalizability.
>
> | Methods             | Dataset | Rel. (%)   | T-Gen. (%) | V-Gen. (%) | T-Loc. (%) | M-Loc. (%) | Avg. (%)   |
> |:--------------------|:--------|-----------:|-----------:|-----------:|-----------:|-----------:|-----------:|
> | VisEdit             | E-VQA   | 97.34     | 96.62     | 96.96     | 100.00    | 84.90    | 95.16     |
> |                     | E-IC    | 95.18     | 95.00     | 94.24     | 100.00    | 93.11     | 95.51     |
> | **DualEdit (Ours)** | E-VQA   | **98.29** | **97.89** | **98.05** | **100.00**| **99.89** | **98.82** |
> |                     | E-IC    | **96.86** | **96.72** | **96.27** | **100.00**| **100.00**| **97.97** |
> >
> > - (**Out-of-Domain Generalizability**).[**$x'$ and $x$ ave largely dissimilar domains.**] To evaluate out-of-domain generalizability, we constructed a benchmark where training and testing data differ significantly, allowing us to better demonstrate the model's generalization capablities.
> >    - (**Benchmark Construction**). Based on the E-VQA dataset, we constructed two completely distinct domains to rigorously evaluate domain generalization. Specifically, for the training set, we retrieved 1,209 samples related to humans by filtering with keywords such as "person," "man," and "woman." In contrast, the test set consists of 218 animal-related samples, selected using keywords like "cat," "dog," and "bird." This deliberate split creates two domains that differ significantly in content and context, allowing us to effectively assess the adapter's ability to generalize from human-related images to an entirely different animal-related domain.
> >     - (**Performance Analysis**). The results shown below indicate that the performance of both methods decreases under the out-of-domain setting. Nevertheless, our method still demonstrates strong generalization capability and consistently outperforms VisEdit, providing further evidence of its robustness.
>
>
> | Methods           | Rel. | T-Gen. | V-Gen. | T-Loc. | M-Loc. | Avg.  |
> |-------------------|------------|--------|--------|--------|--------|-------|
> | VisEdit           | 84.58  | 84.31  | 84.12  | 100.00 | 90.18  | 88.64 |
> | **DualEdit (Ours)** | **86.16** | **85.49** | **85.13** | **100.00** | **100.00** | **91.36** ($\\uparrow$ **2.72**) |

---

> > ### Comment · Reviewer_Ykrx · 2025-06-05
> >
> > Thank you for the authors' thorough response. I am satisfied with how they have addressed most of my questions, particularly regarding the novelty of their work. Based on their clarifications, I would like to increase my review score.

---

> > > ### Comment · Area_Chair_JQmv · 2025-06-05
> > > **Response needed from Ykrx**
> > >
> > > Reviewer Ykrx, please take time to consider the author's response and update your review of the paper.

---

> > > ### Author Response · Authors · 2025-06-07
> > > **We Appreciate Your Feedback**
> > >
> > > Thank you for your positive feedback and for considering an increased review score. We truly appreciate your thoughtful comments and are glad our clarifications addressed your concerns.

---

### Author Response · Authors · 2025-06-04
**To Ac and all Reviewers**

We sincerely thank all reviewers for their valuable and constructive feedback. To facilitate cross-reviewer understanding, we have organized the questions and our corresponding responses in a clear and structured manner. Below, we summarize how we addressed the raised concerns and outline the additional experiments conducted to further strengthen our work.


Clarification of the Details of our Proposed DualEdit
> - (**Finding Novelty, Reviewer Ykrx, Q1**): We clarified that our work primarily focuses on jointly analyzing both textual and visual modalities during the editing process, uncovering modality-specific sensitive layers and distinctly examining the individual impact of each modality.
> - (**Benchmark Preparation, Reviewer Ykrx, Q2**): We explained that we reuse publicly available original-edit paired benchmarks from prior VLM editing works, requiring no extra paired data or more time-consuming preparation than previous methods.
> - (**Theoretical analysis, Reviewer CGGe, Q1**): As theoretical analysis of VLM editing is highly challenging, we instead offer extensive empirical results to support understanding and method design.
> - (**Baseline reproduction differences, Reviewer CGGe, Q3**) All methods, including VisEdit and our DualEdit, were evaluated in a unified experimental environment, using the same hardware, software versions, and evaluation protocols.

Experimental Analysis:
> - (**Generalization, Reviewer Ykrx, Q3**): We provided extensive evaluation on both in-domain and out-of-domain data splits, including a novel human-vs-animal domain benchmark, showing our method’s superior generalization over baselines.
> - (**Necessity of visual editing and its locating, Reviewer DZuA, Q1**): We justified the necessity of the visual editing branch through ablation studies demonstrating significant performance drops without it, and layer-location sensitivity analysis highlighting the importance of inserting the visual editor at the certain layer.
> - (**Soft gating, Reviewer DZuA, Q2**): Following the suggestion, we implemented and tested a soft gating mechanism. While it improves locality, it reduces reliability and generality compared to our binary gating. We explained why our gating mechanism achieves better results.
> - (**Sensitivity to threshold and layer choice, Reviewer DZuA, Q3**): We showed experimentally that the method is robust to threshold settings within a range and that editing at appropriate layers (16–19) yields stable and optimal performance, while editing at very shallow or deep layers harms results.
> - (**Limited datasets and backbones, Reviewer CGGe, Q2**): We clarified that our focus is on Early Projection Fusion VLMs (e.g., LLaVA-V1.5, BLIP2-OPT) due to their architectural characteristics, and that extending to Deep Fusion models (e.g., Flamingo, PaLI) requires different solutions and is out of this paper’s scope.

We have devoted a significant amount of time and effort to address the issues raised in the reviews, and we believe that our rebuttal offers valuable insights that may help resolve some of the concerns. Therefore, we would greatly appreciate it if both the AC and the reviewers could kindly review our responses and consider them during the discussion period.

Thanks again for the time and effort that AC and all the reviewers have put in.

---

### Decision · Program_Chairs · 2025-07-08

**Decision:**

Accept

**Comment:**

Quality: Reviewres agree that the paper contribution is valuable.

Clarity: Reviewers agree that the paper is clear and the methods described seem replicable.

Originality: Reviewers acknowledge that the technical novelty of the work is limited, but that the question under studying (editing VLMs) is worthwhile and underexplored.

Significance: The paper, while not presenting substantial technical novelty, does present significant and actionable findings for the community by introducing and verifying a method for effective VLM editing across multiple backbones.

Pros:
* First comprehensive study decoupling modality-specific impacts in VLM editing, revealing critical insights.
* Provides a much-needed exploration of editing in multimodal models, an area that lags behind LLM editing research.

Cons:
* The conceptual novelty may feel limited, as the work extends existing ideas (like adapter-based editing and modality-specific routing) rather than introducing a fundamentally new editing paradigm.